# Correlation driven near-flat band Stoner excitations in a Kagome magnet

Abhishek Nag [1] ✉, Yiran Peng[2], Jiemin Li [1,3], S. Agrestini[1], H. C. Robarts[1,4], Mirian García-Fernández [1], A. C. Walters [1], Qi Wang[5], Qiangwei Yin[5], Hechang Lei [5], Zhiping Yin [2] ✉ & Ke-Jin Zhou [1] ✉

Among condensed matter systems, Mott insulators exhibit diverse properties that emerge from electronic correlations. In itinerant metals, correlations are usually weak, but can also be enhanced via geometrical confinement of electrons, that manifest as 'flat' dispersionless electronic bands. In the fast developing field of topological materials, which includes Dirac and Weyl semimetals, flat bands are one of the important components that can result in unusual magnetic and transport behaviour. To date, characterisation of flat bands and their magnetism is scarce, hindering the design of novel materials. Here, we investigate the ferromagnetic Kagomé semimetal $Co_3Sn_2S_2$ using resonant inelastic X-ray scattering. Remarkably, nearly non-dispersive Stoner spin excitation peaks are observed, sharply contrasting with the featureless Stoner continuum expected in conventional ferromagnetic metals. Our band structure and dynamic spin susceptibility calculations, and thermal evolution of the excitations, confirm the nearly non-dispersive Stoner excitations as unique signatures of correlations and spin-polarized electronic flat bands in $Co_3Sn_2S_2$. These observations serve as a cornerstone for further exploration of band-induced symmetry-breaking orders in topological materials.

Over the last few decades, there has been an increasing interest in quantum materials with topological electronic band properties[1]. Here, the competition between the Coulomb interaction, Hund's exchange, crystal-field effects, and spin-orbit coupling creates a rich phase space of topologically non-trivial states. These systems including topological insulators, Dirac and Weyl semimetals, display remarkable transport properties when topological bands are tuned to the chemical potential[2–5]. In addition to the highly dispersing Dirac and Weyl bands, another class of bands has also drawn much attention. These bands, flat in momentum space, emerge in special lattices owing to geometrically destructive quantum phase interference of electron hopping pathways. Consequently, the electronic kinetic energy gets suppressed, enhancing electronic interactions. Flat band systems can show diverse many-body phenomena like the fractional quantum Hall effect[6], Mott-insulation[7] and unconventional superconductivity[8]. Combined with magnetism, topological band systems can serve towards energy-efficient spintronics[9], as exemplified by the observation of a giant intrinsic anomalous Hall effect in magnetic Weyl semimetals[10,11].

Among the solid-state systems, Kagomé lattices are the most promising candidates for flat bands. From simple nearest-neighbour tight-binding model considerations, Kagomé lattices may show the presence of both Dirac and flat bands[12]. Moreover, Kagomé materials like twisted bilayer graphene and $Fe_3Sn_2$ have been proposed to host flat band ferromagnetism[6,13,14]. However, the presence of flat bands is not the sole aspect governing bulk magnetic properties in real

[1]Diamond Light Source, Harwell Campus, Didcot OX11 0DE, UK. [2]Department of Physics and Center for Advanced Quantum Studies, Beijing Normal University, 100875 Beijing, China. [3]Beijing National Laboratory for Condensed Matter Physics and Institute of Physics, Chinese Academy of Sciences, 100190 Beijing, China. [4]H. H. Wills Physics Laboratory, University of Bristol, Bristol BS8 1TL, UK. [5]Department of Physics and Beijing Key Laboratory of Opto-Electronic Functional Materials & Micro-Nano Devices, Renmin University of China, 100872 Beijing, China. ✉e-mail: abhishek.nag@diamond.ac.uk; yinzhiping@bnu.edu.cn; kejin.zhou@diamond.ac.uk

materials[14–19]. For example, topological flat bands were revealed in frustrated Kagomé metal CoSn by angle-resolved photoemission spectroscopy (ARPES), however, it is a paramagnet[17,20]. On the other hand, in Kagomé metal $Co_3Sn_2S_2$, although scanning tunnelling spectroscopy (STS)[15,17] has indicated the presence of flat bands in a limited momentum ($q$)-space, it is nevertheless a ferromagnet with a $T_C$ of $\simeq$ 175 K[19]. In fact, $Co_3Sn_2S_2$ is one of the few Kagomé systems that are expected to have flat bands and are also ferromagnets. Also, being a Weyl semimetal, $Co_3Sn_2S_2$ promises tunability of its topological properties via magnetism[10,11,21,22]. To progress the field of band topology-driven magnetism, it is therefore crucial to study the flat band magnetic excitations in these materials. This naturally leads to the search for a technique in complement to ARPES and STS, which is sensitive to not only low-energy collective spin-waves but also band selective magnetic excitations, like resonant inelastic X-ray scattering (RIXS), owing to the presence of an intermediate core-hole state with strong spin-orbit coupling[23].

Here, we investigate the magnetic excitations from flat bands in $Co_3Sn_2S_2$ using Co $L_3$-edge RIXS. We observe a damped excitation peak close to 0.38 eV that has a peak dispersion bandwidth of about 0.05 eV in the probed $q$-space. We calculate the electronic band structure and vertex-corrected dynamic spin structure factor [$S(q, \omega)$] of $Co_3Sn_2S_2$ using a combination of density functional theory and dynamical mean field theory (DFT+DMFT). The calculated $S(q, \omega)$ show an energy ($E$)−$q$-dependence very similar to that in the experiments. Additionally, the comparison of our temperature ($T$)-dependent results to the calculated $S(q, \omega)$ for the ferromagnetic (FM) and paramagnetic (PM) states allows us to associate the experimentally observed peaks to Stoner excitations. We thus identify the nearly non-dispersive Stoner excitation peaks as an unique feature of magnetic excitations from correlated electronic flat bands in $Co_3Sn_2S_2$, in stark contrast to conventional FM metals. Our work also serves as the first experimental $E$−$q$-space evidence for the presence of flat bands in this system.

## Results

In $Co_3Sn_2S_2$, the Co atoms are arranged as stacked layers of Kagomé units as shown in Fig. 1a. It is an itinerant system with a charge carrier density of $1.22 \times 10^{21}$ cm$^{-3}$, and a magnetic moment of 0.2-0.3 $\mu$B/Co, much smaller than that expected from a local moment scenario[19,24]. Comparable resonant energy positions of the Co $L_3$ X-ray absorption (XAS) on $Co_3Sn_2S_2$ with that of Co metal and CoO (Fig. 1b), suggest a $d^7$ electronic configuration of the Co atoms[24]. The spin and orbital momentum of the poorly screened core-hole potential couples with the valence $d$ electrons to give rise to multiplet features in a $L_3$-XAS process, and are indicative of correlation and local crystal-field strengths[25]. While at the strong correlation limit the charge-transfer insulator CoO with Hubbard $U = 5.1$ eV[26], shows multiplet fine structures, at the other extreme, the metallic Co has a featureless asymmetric peak. We note that the XAS of $Co_3Sn_2S_2$ has features qualitatively between the two, indicating an intermediate electronic correlation.

In strongly correlated systems like cuprates or nickelates, excitonic peaks are observed in RIXS from spin, charge, lattice or orbital excitations[23]. Although, strong fluorescence signals from delocalised states usually dominate the RIXS signals in itinerant systems, collective spin-waves have been successfully probed, for instance, in iron-based superconductors or Fe and Ni metals[27–30]. The RIXS response of $Co_3Sn_2S_2$ shown in Fig. 1c, d, also has a strong fluorescence signal linearly dependent on incident energy ($E_i$) above the Co $L_3$-XAS threshold. However, we observe an additional broad but intense peak close to 0.38 eV which remains fairly independent of $E_i$. This shows that the peak results from a coherent excitonic process and provides further proof of moderate electronic correlations despite the itinerancy of the system[18]. At $E_i = 779.85$ eV where the broad peak (S1) at 0.38 eV is most intense, a high-energy resolution RIXS spectrum also reveals a

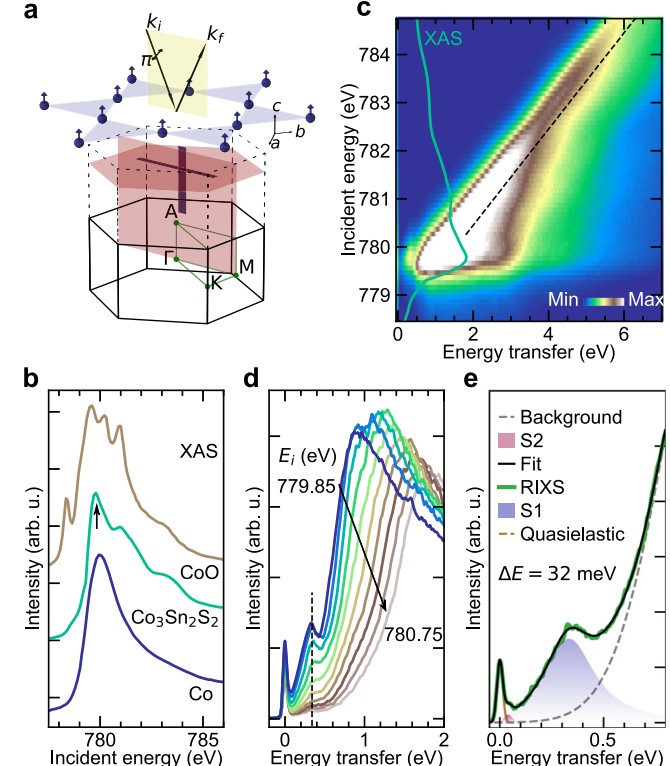

**Fig. 1 | Spectroscopic signatures of correlated states in $Co_3Sn_2S_2$. a** Two dimensional (2D) FM Kagomé unit formed by Co atoms in $Co_3Sn_2S_2$ with the RIXS scattering geometry. The magnetic easy axis is along the crystallographic $c$-direction. RIXS measurements presented have been collected at the points shown by horizontally and vertically shaded strips parallel to Γ-M and Γ-A directions, respectively. **b** Comparison of multiplet features observed in the Co $L_3$-edge XAS of CoO, $Co_3Sn_2S_2$ and Co metal shifted vertically. The arrow shows the $E_i$ used for the momentum and temperature dependent RIXS measurements. **c** RIXS map with $E_i$ varied across the Co $L_3$-edge on $Co_3Sn_2S_2$ at 23 K at (−0.05, 1.60). The dashed line follows the main fluorescence feature. Co $L_3$-edge XAS spectrum of $Co_3Sn_2S_2$ (green solid line) is superimposed on top of the RIXS map. **d** RIXS line cuts from **c** above the Co $L_3$-XAS threshold in steps of $\Delta E_i = 0.1$ eV. The vertical dashed line shows the $E_i$-independent feature. **e** Representative low-energy RIXS spectrum at (0.1, 1.25) with aggregated least-square fit of different low-energy peak profiles and a high energy background. As discussed in the text, S1 and S2 primarily represent Stoner excitations from spin-polarised flat bands.

low-energy peak (S2) close to 0.04 eV. Spin-wave excitations in $Co_3Sn_2S_2$ have been observed using inelastic neutron scattering (INS) up to 0.018 eV[31], and as such cannot be resolved from the elastic peak at zero energy with the present energy resolution.

In Fig. 2a, b we present the spin-resolved $d$-orbital electronic bands in the FM state calculated with $U = 5.0$ eV and Hund's exchange $J_H = 0.9$ eV, showing the presence of dispersing as well as flat bands in $Co_3Sn_2S_2$. The overall low-energy band dispersions and energies are in good agreement with reported ARPES data on $Co_3Sn_2S_2$ (see "Methods")[22,32]. Along with the spin-waves, electron–hole pair excitations with electrons and holes in the bands of opposite spins are also elementary to itinerant ferromagnets, and are called the Stoner excitations. The distribution of Stoner excitations in the $E − q$-space, however, is strongly influenced by the band structure. In conventional ferromagnetic metals with highly dispersing bands, the Stoner excitations are typically spread over several eVs as a continuum (Fig. 2c, d). The continuum presents no peak-like feature in INS or RIXS experiments and manifests as damping or renormalisation of spin-wave energies for overlapping $E − q$ values[29]. While in the itinerant flat band ferromagnets, the spin-wave energies are expected to maintain qualitatively similar dispersion behaviour (albeit with possible energy

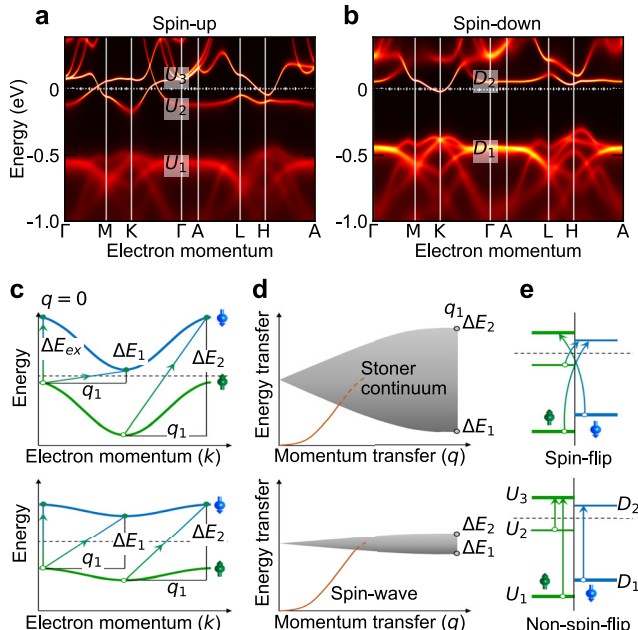

**Fig. 2 | Flat band Stoner excitations in $Co_3Sn_2S_2$.** Calculated spin-resolved electronic bands in $Co_3Sn_2S_2$ for the FM state with (**a**) spin-up and (**b**) spin-down bands. **c** Schematic representation of electronic bands and spin-flip Stoner excitations comprising of electron–hole pairs of opposite spins, shown for zero ($\Delta k = q = 0$) and non-zero ($\Delta k = q_1 \neq 0$) momentum transfer values. For a rigid band splitting, $q = 0$ excitations have an energy equal to the exchange splitting as seen in (**c**). For non-zero values of $q$, excitation pathways with different energies are satisfied giving rise to a continuum of excitations called the Stoner continuum spread across the $E$–$q$-space ($\Delta E_2 \gg \Delta E_1$). **d** Stoner continuum of excitations corresponding to the electronic bands shown in (**c**). As the bands become flatter or dispersionless, the range of energies for these pathways reduces giving rise to a narrower band of excitations ($\Delta E_2 \simeq \Delta E_1$). **e**, Schematic spin-flip and non-spin-flip $q = 0$ excitations close to $\Gamma$, from and to the points marked in (**a, b**).

renormalisation) as in ordinary itinerant ferromagnets[33–35], flattening of the bands confine the spectral weight distribution of the Stoner continuum to a narrow region of the $E$–$q$ space (Fig. 2c, d)[33–35]. Stoner excitations across segments of flat bands as shown in Fig. 2a, e[23], may appear in Co L-edge RIXS and give rise to the S1 peak in $Co_3Sn_2S_2$, and therefore, next we investigate its dependence in the $E$–$q$-space.

Figure 3a, b shows the low-energy RIXS spectra from $Co_3Sn_2S_2$ along the $\overline{\Gamma M}$ and $\overline{\Gamma A}$ directions. Panels c and d show the corresponding intensity maps with the quasielastic and background contributions subtracted, thereby, highlighting the S1 and S2 peaks. The S1 peaks were fitted with a generic damped harmonic oscillator model having undamped peak energy $\omega_0$ and peak damping factor $\gamma/\omega_0$ (Fig. 1e and "Methods"). The largest value of $\omega_0$ for S1 is found to be 0.4 eV. The $\omega_0$s of the S2 peaks were kept fixed at 0.04 eV as extracted from high energy resolution RIXS spectra (see "Methods" and Supplementary Information Fig. S5 and S6 showing the non-dispersing S2 peaks). The effect of correlations in the two-particle response functions are generally taken into account following a vertex correction[36]. For example, the dynamic spin susceptibility [$\chi(q,\omega)$] is related to the bare spin susceptibility [$\chi_0(q,\omega)$] as $\chi(q,\omega) = \chi_0(q,\omega)/[1-\Gamma_V(\omega)\chi_0(q,\omega)]$, where $\Gamma_V$ is the two-particle vertex function. For weak or uncorrelated itinerant systems, the dynamic spin susceptibility resembles the bare susceptibility owing to a small or zero vertex correction. For correlated systems, a significant renormalisation of spectral shape occurs as $\Gamma_V(\omega)$ Re $\chi_0(q,\omega)$ values approach 1 giving rise to the poles in $\chi(q,\omega)$. We calculated the bare spin and charge susceptibilities, i.e., the electron–hole pair excitations, based on the DFT+DMFT band structure results with the inclusion of the screened Coulomb interaction and found that both the bare susceptibilities are featureless in the

energy range of S1, and have no spectral weight in the energy range of S2 (see Methods and Supplementary Information Fig. S7). Figure 3e, f show the dynamic structure factor $S(q,\omega) = \text{Im } \chi(q,\omega)/(1 - e^{-\hbar\omega/k_B T})$, obtained after vertex correction (see "Methods"). The remarkable similarity in the energy scales and the dispersions of S1 (S2) peaks between the experiment and the calculated $S(q,\omega)$, suggests their origin from the spin-flip Stoner excitations in $Co_3Sn_2S_2$, owing to schematically shown $U_1 \rightarrow D_2$ or $D_1 \rightarrow U_3$ ($U_2 \rightarrow D_2$)-like transitions in Fig. 2e. At the same time, the significant difference between bare and vertex corrected susceptibilities (see Supplementary Information Fig. S7), underlines the importance of the correlations effects in both the ground state and the collective excitations. We also note that the experimental distributions are distinct from the vertex corrected dynamic charge susceptibilities representing non-spin-flip charge excitations, as shown in Fig. 3g, h. In optical studies of $Co_3Sn_2S_2$, a peak close to 0.2 eV was observed in the FM state, which gradually broadened in the PM state and was assigned to non-spin-flip excitations from the flat bands[16]. Our charge susceptibility calculations are consistent with the optical result. While spin-conserved processes are also allowed in RIXS, we do not observe any additional peak-like feature in the almost linear RIXS spectra between 0.1 – 0.38 eV[16]. We thus refrain from performing a multi-component fitting of S1 and report the single component $\omega_0$ and $\gamma/\omega_0$ in this work. The potential contribution of the charge susceptibility could be the reason behind the large damping ($\gamma/\omega_0 \approx 0.5$), observed for S1 compared to the calculations. Similarly, while the energy scale of S2 matches with the low-energy component of calculated spin-susceptibility, we cannot discard optical phonon contributions at this energy[18]. A quantitative estimation of these contributions to the effective RIXS scattering cross-section is however, beyond the capabilities of presently available computational methods.

To further demonstrate the dominant contribution of Stoner excitations to the S1 peaks, we collected temperature-dependent RIXS spectra across the $T_C$ of $Co_3Sn_2S_2$. As shown in Fig. 4a, the peak height of S1 with $\omega_0 \simeq 0.38$ eV, decreases till the $T_C$, above which it becomes featureless and shows almost no change till 347 K in the PM state. In Fig. 4b, we present the temperature-dependent RIXS intensity map along with $\omega_0$ of S1 extracted from fitting. We observe clearly the spectral weight shift along with the change in $\omega_0$ below and above $T_C$. Such a shift and loss in the peak height, seen in Fig. 4a, is expected because the bands become partially non-spin-polarised and shift in energy above $T_C$ (see Supplementary information Fig. S8)[32]. This is also reflected in the concomitant shift towards higher energy in the calculated $S(q,\omega)$ in the PM state (Fig. 4c and Supplementary Information Fig. S8). Similarly, for paramagnetic CoSn which does not have spin-polarised bands, Co $L_3$-edge RIXS collected at 20 K is devoid of any peak-like feature in this region, as shown in Fig. 4a (see also Supplementary Information Fig. S9). It was shown in Ref. [19], that $Co_3Sn_2S_2$ exhibits a c-axis FM ground state until $T_C^* \approx 90$ K, above which an additional ab-plane antiferromagnetic component starts growing, eventually attaining a volume fraction of 80% at $\simeq 170$ K and finally becoming PM above $T_C$. From Fig. 4b, d, we observe that the $\omega_0$ and $\gamma/\omega_0$ remain nearly constant below $T_C^*$, and starts rising linearly with the reduction in the FM volume fraction. Above $T_C$, when the system becomes PM, $\omega_0$ again stops changing, along with the decline in the rate of change of damping. In itinerant ferromagnets the band structure evolves continuously with temperature and our results show that the observed excitations are intricately linked to the magnetism in $Co_3Sn_2S_2$. Since we observe spectral weights, although largely damped, till almost twice the $T_C$, this indicates that the magnetic moment fluctuations and electronic correlations persist much above $T_C$, also suggested by Rossi et al. in their recent work on $Co_3Sn_2S_2$[37].

## Discussions

While $Co_3Sn_2S_2$ has attracted a lot of interest in recent years due to its topological properties and magnetism, there has been no momentum-

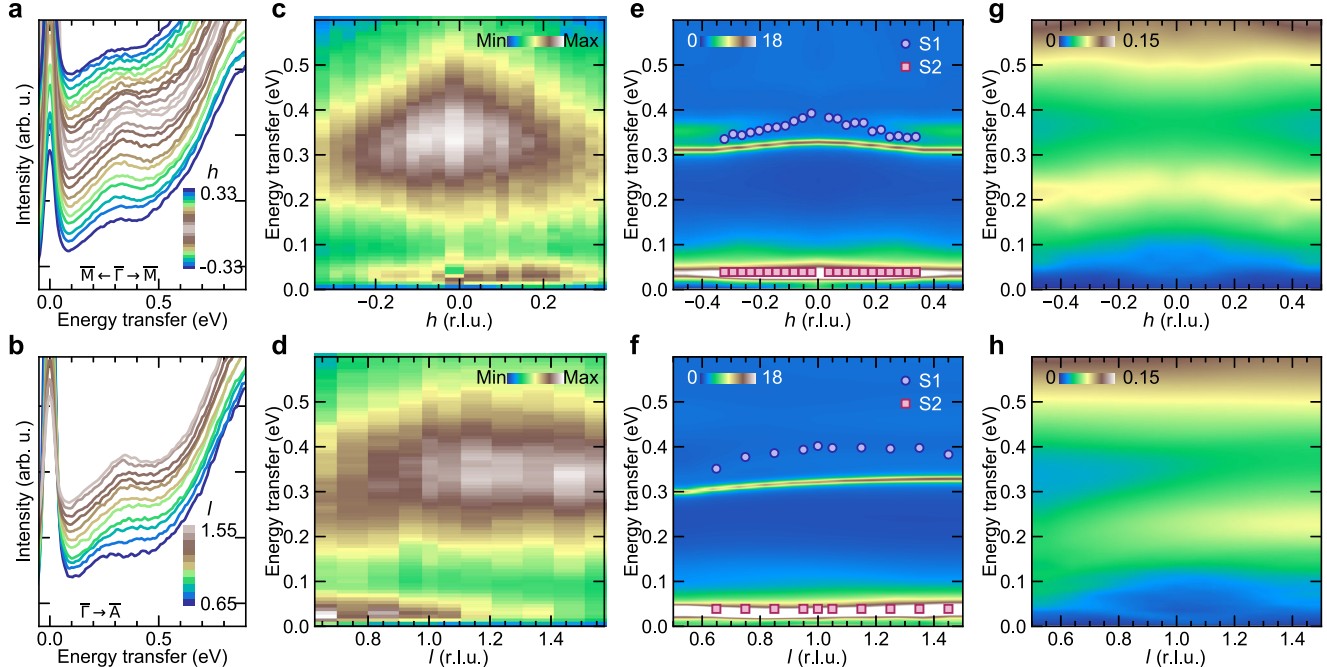

**Fig. 3 | Dispersion of the flat band Stoner excitations in the $E − q$-space in Co$_3$Sn$_2$S$_2$.** RIXS line spectra for momentum transfer parallel to (**a**) $\overline{\Gamma M}$ direction from $h = −0.33$ to $h = 0.33$ at $l = 1.25$ and **B**, $\overline{\Gamma A}$ direction from $l = 0.65$ to $l = 1.55$ at $h = 0.05$. **c**, **d** RIXS intensity maps corresponding to the line spectra shown in (**a**, **b**), with fitted elastic and background contributions subtracted. **e**, **f** Calculated vertex corrected dynamic spin structure factor $S(q, \omega)$ intensity maps along the RIXS

measurement directions for spin-flip Stoner excitations. The markers are the $\omega_0$ values of S1 and S2 peaks extracted from least-square fits of RIXS spectra (see "Methods"). **g**, **h** Calculated vertex corrected dynamic charge structure factor intensity maps along the RIXS measurement directions for non-spin-flip charge excitations.

resolved experimental evidence of flat bands in this system[15,18]. We have demonstrated with the observation of Stoner excitation peaks with a dispersion bandwidth of about 0.05 eV over 65% of a Brillouin zone in the $\overline{\Gamma M}$ direction, the presence of flat bands in this system. It should be highlighted that the dispersion bandwidth estimated from the Stoner excitations has the combined contribution of both the valence and conduction bands, as well as the $q$-dependent exchange splitting energy, and therefore can serve as a more stringent test for the existence of flat bands. The Stoner excitation peaks also have a dispersion bandwidth of about 0.05 eV along the out-of-plane direction suggesting the presence of interlayer electron interactions and deviation from ideal two dimensionality in this system[38].

The excitations observed in the experiments are not particle-hole excitations from the joint density of states as demonstrated by the absence of peak-like features in the calculated bare spin/charge susceptibilities at the energy scales of S1 and S2 peaks. The resemblance in the energy scales and dispersion between the experiments and the vertex-corrected dynamic spin susceptibilities show the importance of correlations driving the two-particle spin response or the Stoner excitations from the flat bands in this system. Furthermore, the excitations are intricately linked to the thermal evolution of the electronic bands and the associated magnetism.

The low-energy excitations are crucial to understand the interaction-driven many-body ground states governing the material properties when the bands are tuned to the chemical potential. While ARPES is the choice tool to investigate the valence bands our results demonstrate that RIXS can be utilised for identifying the presence of flat bands and the energy scales in both valence and conduction bands, providing information that may be utilised to modify the electronic/ magnetic properties of these materials by doping. Direct experimental observation of Stoner excitations in RIXS also means that it can be used to clarify the magnon-Stoner interactions in itinerant correlated flat band systems like FeSn where INS have observed discrepancies in magnon intensities at energies relevant to Stoner excitations[39].

## Methods

### Sample details

Co$_3$Sn$_2$S$_2$ single crystals were grown by the Sn flux method. The starting elements of Co (99.99 %), Sn (99.99 %) and S (99.99 %) were put into an alumina crucible, with a molar ratio of Co:S:Sn = 4:3:43 The mixture was sealed in a quartz ampoule under partial argon atmosphere and heated up to 1323 K, then cooled down to 973 K at 5 K/h. The Co$_3$Sn$_2$S$_2$ single crystals were separated from the Sn flux by using centrifuge. FM transition temperature $T_C$ from magnetisation measurements was found to be 172 K. See Supplementary Information for sample characterisation details.

### Theoretical calculations

A combination of density functional theory and dynamical mean field theory (DFT+DMFT)[40] was used to compute the electronic band structure of Co$_3$Sn$_2$S$_2$ in the FM and PM states (see Supplementary information Fig. S10). The full-potential linear augmented plane wave method implemented in Wien2K[41] was used for the DFT part. The Perdew-Burke-Ernzerhof type generalised gradient approximation[42] was used for the exchange-correlation functional. DFT+DMFT was implemented on top of Wien2K which is described in detail in ref. [43]. In the DFT+DMFT calculations, the electronic charge was computed self-consistently on DFT+DMFT density matrix. The quantum impurity problem was solved by the continuous time quantum Monte Carlo (CTQMC) method[44,45] with $U = 5.0$ eV and Hund's exchange $J_H = 0.9$ eV in both FM and PM states. Bethe-Salpeter equation was used to compute the dynamic spin susceptibility where the bare susceptibility was computed using the converged DFT+DMFT Green's function, while the two-particle vertex was directly sampled using CTQMC method after achieving full self-consistency of DFT+DMFT density matrix[36]. For the FM state, the averaged Green's function of the spin-up and spin-down channels was used to compute the bare susceptibility. The experimental crystal structure (space group $R\bar{3}m$, No. 166) of Co$_3$Sn$_2$S$_2$ with

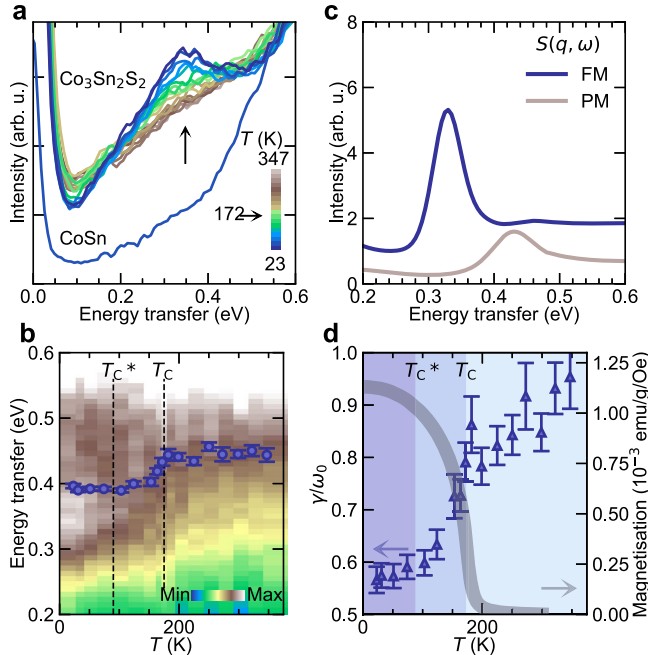

**Fig. 4 | Temperature dependence of the flat band Stoner excitations in Co₃Sn₂S₂.** Low-energy RIXS spectra at $(0.025, 1.25)$ at temperatures ranging from 23 K to 347 K shown as (**a**) lines and (**b**) intensity map for $Co_3Sn_2S_2$. Also compared in (**a**) is the low-energy RIXS spectra from paramagnetic flat band material CoSn at $(-0.15, 0.5)$ collected at 20 K. The markers in (**b**) are the $\omega_0$ values of S1 extracted from least-square fits of RIXS spectra. **c** Calculated $S(q, \omega)$ convoluted with the experimental resolution for the FM and the PM band configurations at $(0.025, 1.25)$. **d** Extracted damping of S1 as a variation of temperature compared to the magnetisation data of $Co_3Sn_2S_2$. Shaded regions from 0–90 K, 90–175 K and 175–350 K represent c-axis FM, c-axis FM with ab-plane antiferromagnetic, and the PM magnetisation volume, respectively, in $Co_3Sn_2S_2$ under zero-applied-field from Ref. [19] Error bars are least-square-fit errors.

hexagonal lattice constants $a = b = 5.3689$ Å and $c = 13.176$ Å was used in the calculations.

## RIXS measurements

The Co $L_3$-edge XAS spectra shown in Fig. 1b were collected in total electron yield mode using $\sigma$-polarised X-ray. Momentum transfer $q = ha^* + kb^* + lc^*$ is defined using Miller indices $(h, k, l)$ in reciprocal lattice units. We state the values of $(h, l)$ and $k = 0$ if not stated explicitly. Crystals were cleaved in vacuum and the pressure in the experimental chamber was maintained below $5 \times 10^{-10}$ mbar. RIXS spectra presented at Co $L_3$-edge were collected with an energy resolution of $\triangle E \simeq 0.048$ eV, at I21-RIXS beam line, Diamond Light Source, United Kingdom[46]. Additional spectra were also collected with $\triangle E \simeq 0.032$ eV. Samples were mounted such that the c-axis was in the horizontal scattering plane (Fig. 1a). While the $E_i$-dependent RIXS spectra were collected using $\pi$-polarised X-ray, the $q$ and $T$-dependent RIXS spectra were collected using $\sigma$-polarised X-ray. The zero-energy transfer position and resolution of the RIXS spectra were determined from subsequent measurements of elastic peaks from an amorphous carbon sample.

## RIXS data fitting

RIXS data were normalised to the incident photon flux, and subsequently corrected for self-absorption effects, prior to fitting. A Gaussian lineshape with the experimental energy resolution was used to fit the quasielastic line. A high-energy Gaussian background was included in the fitting model to account for the contribution of fluorescence features. The scattering intensities $S(\mathbf{q}, \omega)$ of peaks S1

and S2 were modelled using a generic damped harmonic oscillator function:

$$S(\mathbf{q}, \omega) \propto \frac{1}{1 - e^{-\hbar\omega/k_B T}} \frac{\gamma\omega}{[\omega^2 - \omega_0^2]^2 + 4\omega^2\gamma^2}, \quad (1)$$

where $k_B$, $T$ and $\hbar$ are the Boltzmann constant, temperature and the reduced Planck constant. $\omega_0$ and $\gamma$ are the undamped frequency and the damping factor of the peaks, respectively. The peak is underdamped if $\gamma/\omega_0 < 1$. See Supplementary Information Fig. S2-S4 showing the fitting profiles. Since the S2 peak cannot be resolved from the quasielastic peak in the spectra with $\triangle E \simeq 0.048$ eV, the S2 peak positions (0.04 eV) were determined from the spectra collected with $\triangle E \simeq 0.032$ eV and fixed for the lower resolution scans (Supplementary Information Figs. S5, S6).

## Data availability

The experimental data presented in the figures are available as a public data set at https://doi.org/10.5281/zenodo.7264820.

## Code availability

The codes used for the DFT+DMFT calculations in this study are available from the corresponding authors upon reasonable request.

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

## Acknowledgements

The authors thank R. Comin, H. Luo, H. Weng, and E. Liu for insightful discussions. We also thank Y. Huang for the support of the samples. All data were taken at the I21 RIXS beamline of Diamond Light Source (United Kingdom) using the RIXS spectrometer designed, built and owned by Diamond Light Source. We thank Diamond Light Source for providing beam time under proposal ID NR27905. We acknowledge T. Rice for the technical support throughout the experiments. We also thank G. B. G. Stenning and D. W. Nye for help on the Laue instrument in the Materials Characterisation Laboratory at the ISIS Neutron and Muon Source. Z.Y. was supported by the NSFC (Grant Nos. 12074041 and 11674030) and the National Key Research and Development Program of China grant 2016YFA0302300. The calculations used high performance computing clusters at Beijing Normal University, China. H.C.L. was supported by National Key R&D Program of China (Grant Nos. 2018YFE0202600 and 2022YFA1403800), Beijing Natural Science Foundation (Grant No. Z200005), National Natural Science Foundation of China (Grant No. 12274459).

## Author contributions

K.-J.Z. conceived the project. K.-J.Z. and A.N. supervised the project. A.N., K.-J.Z., J.L., S.A., H.C.R, M.G.-F., and A.C.W. performed RIXS measurements. A.N. and K.-J.Z. analysed RIXS data. Q.W., Q.W.Y., and H.C.L. synthesised and characterised the sample. Y.P. and Z.Y. performed DMFT calculations. A.N. and K.-J.Z. wrote the manuscript with comments from all the authors.

## Competing interests
