## [Peer Review File · Nature Communications]

Reviewers' Comments:

Reviewer #1:

Remarks to the Author:

The authors have addressed most of my concerns. Perhaps the paper can now be accepted for publication in *Nature Communications*.

Reviewer #2:

Remarks to the Author:

I have carefully read the new manuscript by Nag et al, now titled "Correlation driven near-flat band Stoner excitations in a Kagome magnet". The authors have addressed almost all the issues I raised when reviewing the previous version of this manuscript. Still, I have a few important questions/comments that require some clarification. If clarified, I am supportive of publication in *Nature Communications*.

The main observation is made by E-q mapping with RIXS. Two features (S1 and S2) are observed and characterized. The authors use both bare and vertex-corrected calculations of the charge and spin susceptibility to identify the origin of these features. Correctly, they start with the simplest calculation. However, the bare susceptibility calculation does not yield spectral weight or features that would account for the experiments - this is clear. Next they use a vertex correction calculation for the susceptibility. This calculation does a much better job at capturing the most salient features of the experimental data. The authors then reach 2 conclusions: 1) the S1 feature is from Stoner excitations from flat bands and 2) Correlations are necessary to explain the experimental data.

In more detail (I apologize that I am re-writing the authors arguments, but I wanted to be clear regarding the logic I am using to get to my questions).:

1 - Regarding the Stoner excitations from flat bands: I tend to agree with the authors argument that if they are seeing something as well-defined (in the sense that authors used in the previous version) it must be indicating the flat band above the Fermi-level as well. Whether it is a spin-flip transition requires comparison to calculations (as acknowledged by the authors). They provide 4 calculations, 2 of charge and 2 of spin susceptibilities. Only one of them gets close to explaining S1 (the spin susceptibility with correlations). However, believing that this is evidence of spin-flip scattering requires that we believe the calculation of the band structure.

— Question: Is the calculation in agreement with actual reality? Is the structure in the calculations consistent with ARPES, for the occupied side? We need to have some level of confidence in this. The authors should make this important comment or statement in the text.

2 - Correlation effects being required. I generally understand, as an experimentalist, how the vertex corrected calculations and DMFT include correlations. However, I am a bit surprised at how much it changed the calculation of the susceptibilities. Could the authors provide a simple outline for why this is the case? As it is, the manuscript just states that correlations should modify $S(q,w)$ and that DMFT can capture this. But this is impenetrable to a broad audience. A few more sentences trying to provide a physical intuition would be welcome.

Smaller comments

Fig. 2c: Difficult to see the direction of the arrow.

Fig. 2 caption. The letter d in bold is missing to indicate the description of that panel.

Lines 105-110 took me several readings. The first sentence says there are no similarities between the bare susceptibility and the data. The next sentence says there is a "remarkable similarity". The next sentence refers to a "striking difference". I'm sure the authors can see how it reads in a confusing manner. Please clarify.

POINT-BY-POINT REPLY TO PEER REVIEW COMMENTS

RESPONSE TO REVIEWER #1

Reviewer: The authors have addressed most of my concerns. Perhaps the paper can now be accepted for publication in Nature Communications.

Our response: We thank the referee for their re-reading of our manuscript, and are pleased that they find the resubmitted version of the paper suitable for publication in Nature Communications.

RESPONSE TO REVIEWER #2

Reviewer: I have carefully read the new manuscript by Nag et al, now titled “Correlation driven near-flat band Stoner excitations in a Kagome magnet”. The authors have addressed almost all the issues I raised when reviewing the previous version of this manuscript. Still, I have a few important questions/comments that require some clarification. If clarified, I am supportive of publication in Nature Communications.

Our response: We thank the referee for their re-reading of our manuscript, and are pleased that we could incorporate and address their comments and suggestions in our resubmitted version.

Reviewer: The main observation is made by E-q mapping with RIXS. Two features (S1 and S2) are observed and characterized. The authors use both bare and vertex-corrected calculations of the charge and spin susceptibility to identify the origin of these features. Correctly, they start with the simplest calculation. However, the bare susceptibility calculation does not yield spectral weight or features that would account for the experiments - this is clear. Next they use a vertex correction calculation for the susceptibility. This calculation does a much better job at capturing the most salient features of the experimental data. The authors then reach 2 conclusions: 1) the S1 feature is from Stoner excitations from flat bands and 2) Correlations are necessary to explain the experimental data.

In more detail (I apologize that I am re-writing the authors arguments, but I wanted to be clear regarding the logic I am using to get to my questions):

1 - Regarding the Stoner excitations from flat bands: I tend to agree with the authors argument that if they are seeing something as well-defined (in the sense that authors used in the previous version) it must be indicating the flat band above the Fermi-level as well. Whether it is a spin-flip transition requires comparison to calculations (as acknowledged by the authors). They provide 4 calculations, 2 of charge and 2 of spin susceptibilities. Only one of them gets close to explaining S1 (the spin susceptibility with correlations). However, believing that this is evidence of spin-flip scattering requires that we believe the calculation of the band structure.

— Question: Is the calculation in agreement with actual reality? Is the structure in the calculations consistent with ARPES, for the occupied side? We need to have some level of confidence in this. The authors should make this important comment or statement in the text.

Our response: In Fig. R1, we present a comparison of the available ARPES results and our band structure calculations obtained from a combination of DFT+DMFT. It can be seen from the ARPES intensity plots [Fig. R1(a, d)] and our calculated band structure [Fig. R1(b, e)] of $\text{Co}_3\text{Sn}_2\text{S}_2$ that, the overall dispersions and energy scales are in good agreement. It should be noted that while the calculations are at the exact high symmetry points in the Brillouin zone, the experiments probe different values of k_z along the projected high symmetry paths. For example, the gray curve in

Fig. R1c shows the variation of k_z for incident photon energy of 115 eV. Additionally, while DFT based band structure needs to be bandwidth renormalised and shifted to match the ARPES data (see red curves in Fig. R1d), our DFT+DMFT calculations show band structure having similar energy scales as experiment without any renormalisation. We have now included a sentence in the manuscript stating that our band structure calculations are in good agreement with reported ARPES results.

Reviewer: 2 - Correlation effects being required. I generally understand, as an experimentalist, how the vertex corrected calculations and DMFT include correlations. However, I am a bit surprised at how much it changed the calculation of the susceptibilities. Could the authors provide a simple outline for why this is the case? As it is, the manuscript just states that correlations should modify $S(q,\omega)$ and that DMFT can capture this. But this is impenetrable to a broad audience. A few more sentences trying to provide a physical intuition would be welcome.

As the referee mentions, it is important to include the vertex corrections to obtain the two-particle dynamic susceptibilities of correlated electronic systems. The dynamic spin susceptibility $[S(q, \omega)]$ is related to the bare spin susceptibility $[S_0(q, \omega)]$ as $S(q, \omega) = S_0(q, \omega) / [1 - \Gamma_V(\omega) S_0(q, \omega)]$, where $\Gamma_V(\omega)$ is the two-particle vertex function. For weak or uncorrelated metals, the dynamic spin susceptibility resembles the bare susceptibility owing to a small or zero vertex correction. For correlated systems, a significant renormalisation of spectral shape occurs as $\Gamma_V(\omega) S_0(q, \omega)$ values approach 1 giving rise to the poles in $S(q, \omega)$. Following the referee's suggestion to include this description, we have now included a few sentences in the manuscript.

Reviewer: Smaller comments Fig. 2c: Difficult to see the direction of the arrow.

Our response: We have changed the position of the arrow heads, so the arrow directions are now clearer.

Reviewer: Fig. 2 caption. The letter d in bold is missing to indicate the description of that panel.

Our response: We have added now the description of panel d in Fig. 2 caption.

Lines 105-110 took me several readings. The first sentence says there are no similarities between the bare susceptibility and the data. The next sentence says there is a "remarkable similarity". The next sentence refers to a "striking difference". I'm sure the authors can see how it reads in a confusing manner. Please clarify.

Our response: We thank the referee for this suggestion. We have now rewritten this section to clarify the effects of correlations, vertex corrections and comparison of theory and experimental results.

List of changes

All changes in the main text have been highlighted in red in the new version of our manuscript.

1. We have added the following text in line 80: **The overall low-energy band dispersions and energies are in good agreement with ARPES data on $\text{Co}_3\text{Sn}_2\text{S}_2$.**
2. We have added in the caption of Fig. 2 of main text, the description of panel d: **d, Stoner continuum of excitations corresponding to the electronic bands shown in panel c.**
3. We have rewritten the section describing the vertex correction and comparison of experimental results and theory following the suggestions of the referee. **The effect of correlations in the two-particle response functions are generally taken into account following a vertex correction [35]. For example, the dynamic spin susceptibility $[S(q, \omega)]$ is related to the bare spin susceptibility $[S_0(q, \omega)]$ as $S(q, \omega) = S_0(q, \omega)/[1 - \Gamma_V(\omega)S_0(q, \omega)]$, where Γ_V is the two-particle vertex function. For weak or uncorrelated itinerant systems, the dynamic spin susceptibility resembles the bare susceptibility owing to a small or zero vertex correction. For correlated systems, a significant renormalisation of spectral shape occurs as $\Gamma_V(\omega)S_0(q, \omega)$ values approach 1 giving rise to the poles in $S(q, \omega)$. We calculated the bare spin and charge susceptibilities, *i.e.*, the electron-hole pair excitations, based on the DFT+DMFT band structure results with the inclusion of the screened Coulomb interaction and found that both the bare susceptibilities are featureless in the energy range of S1, and have no spectral weight in the energy range of S2 (see Methods and Supplementary Information Fig. S7). Fig.3e and f show the $S(q, \omega)$, obtained after vertex correction (see Methods). The remarkable similarity in the energy scales and the dispersions of S1 (S2) peaks between the experiment and the calculated $S(q, \omega)$, suggests their origin from the Stoner excitations in $\text{Co}_3\text{Sn}_2\text{S}_2$, owing to schematically shown $U_1 \rightarrow D_2$ or $D_1 \rightarrow U_3$ ($U_2 \rightarrow D_2$)-like transitions in Fig 2e. At the same time, the significant difference between $S(q, \omega)$ and $S_0(q, \omega)$ (see Supplementary Information Fig. S7), underlines the importance of the correlations effects in both the ground state and the collective excitations. We also note that the experimental distributions are distinct to those of the vertex corrected charge susceptibilities representing non-spin-flip excitations as shown in Fig.3g and h.**
4. We have changed the position of the arrow heads, so that the arrow directions are clearer in Fig 2c.

Figure R1: Comparison of ARPES data on $\text{Co}_3\text{Sn}_2\text{S}_2$ with band structure calculations.
a, ARPES intensity plot along $\bar{\Gamma}-\bar{K}-\bar{M}'-\bar{\Gamma}-\bar{M}$ from Liu, D. F. et al., *Phys. Rev. B* **104**, 205140 (2021).
b, Calculated electronic bands for the ferromagnetic state along $\Gamma-K-M'-\Gamma-M$ using DFT+DMFT in the present work.
c, ARPES intensity plot along k_y-k_z plane with energy integration window from $E_F-0.1$ eV to E_F from Liu, D. F. et al., *Science* **365**, 1282–1285 (2019). The black curve indicates the k_z momentum locations probed by 115-eV photons. The dashed line marked as ‘cut1’ indicates the momentum direction $\bar{M}_1-\bar{M}_2-\bar{M}_3$ of the ARPES data shown in panel **d**. The red curves in panel **d** are from DFT calculations, where the calculated bandwidth was renormalised by a factor of 1.43 and the energy position was shifted to match the experiment.
e Calculated electronic bands for the ferromagnetic state along M_1-M_2 using DFT+DMFT in the present work.

Reviewers' Comments:

Reviewer #2:

Remarks to the Author:

The authors have addressed all my comments and I recommend publication of the work in Nature Communications. My last suggestion is that the authors also include the ARPES-theory comparison figure (R1 in the rebuttal) to the supplemental information section.

POINT-BY-POINT REPLY TO PEER REVIEW COMMENTS

RESPONSE TO REVIEWER #2

Reviewer: The authors have addressed all my comments and I recommend publication of the work in Nature Communications. My last suggestion is that the authors also include the ARPES-theory comparison figure (R1 in the rebuttal) to the supplemental information section.

Our response: We thank the referee for their re-reading of our manuscript, and are pleased that they find the resubmitted version of the paper suitable for publication in Nature Communications. We have now included as per the suggestion of the referee to the ARPES-theory comparison figure R1 to the supplemental information section.

List of changes

All changes in the main text have been highlighted in red in the new version of our manuscript.

1. We have now included as per the suggestion of the referee to the ARPES-theory comparison figure R1 to the supplemental information section.
2. We have made minor typographical changes in the main text highlighted in red.